# The Local Rhombus-Shaped Flap—An Easy and Reliable Technique for Oncoplastic Breast Cancer Surgery and Defect Closure in Breast and Axilla

**DOI:** 10.3390/cancers16173101

**Published:** 2024-09-06

**Authors:** Hisham Fansa, Sora Linder

**Affiliations:** Department of Plastic Surgery and Breast Center Zürich, Spital Zollikerberg, Zollikerberg, 8125 Zurich, Switzerland

**Keywords:** breast cancer, breast-conserving therapy, oncoplastic surgery, extreme oncoplasty, local flap, Limberg flap, rhombus-shaped flap, lateral breast, volume replacement, volume displacement

## Abstract

**Simple Summary:**

Breast-conserving therapy to resect only the tumor in breast cancer is a standard procedure. To replace loss of volume in the breast and to avoid dislocation of the nipple/areola, local flaps are useful. One technique that has only rarely been used is the local, pedicled rhombus-shaped flap, according to Limberg. The flap is easy and safe, and the donor site is always closed primarily. On the breast, the rhombus-shaped flap can be used as an all-layer flap to cover defects, or it can be deepithelialized to be used as a buried flap. The flap can be harvested from the breast itself or the chest wall. Especially in the lateral breast, the flap is very useful. In the axilla, it can be used to cover full-thickness defects after resection when skin is involved, avoiding a contracture in the axilla.

**Abstract:**

Primarily, breast-conserving therapy is an oncological intervention, but eventually it is judged by its cosmetic result. Remaining cavities from tumor resection can promote seromas, delay healing and cause lasting discomfort. Additionally, volume loss, dislocation of nipple/areola and fat necrosis lead to (cosmetically) unfavorable results, aggravated by radiotherapy. Oncoplastic surgery can reduce these sequelae. A local flap that has rarely been used in breast cancer surgery is the Limberg rhombic flap. The tumor defect is planned as a rhombus. The sides of the rhombus are of equal length and ideally have an angle of 60° and 120°. The flap that closes the defect is planned as an extension of equal length of the short diagonal. The second incision of the flap is placed according to the defect angle of 60°, running parallel to the defect at the same length. This creates a second rhombus. The flap is transposed into the defect, and the donor area is primarily closed. It is axially perfused and safe with a 1:1 length-to-width ratio. Compared to local perforator flaps, defect closure is easily managed without microsurgical skills. In the breast, the flap can be used in volume replacement and volume displacement techniques as an all-layer flap to cover defects, or it can be deepithelialized and buried. In the axilla, it can cover full-thickness defects when skin is involved. The advantages of the rhombic flap are its safety and simplicity to add volume and close defects, thus reducing the complexity of oncoplastic surgery.

## 1. Introduction

Breast-conserving therapy (BCT) for breast cancer is a standard procedure and is oncologically safe given that the indications are correct [1]. As a rule, the parenchymal defects in the breast resulting from tumor resection are closed by local breast parenchymal flaps in order to leave no cavities. Cavities promote seromas, delay healing and the follow-up treatment and can cause lasting discomfort, especially after radiotherapy and radiotherapy with a boost [2,3,4]. They also make aftercare more difficult. Tension and reduced blood flow can lead to fat necrosis if parenchymal flaps are inadequate in size or volume or are inadequately perfused.

With an unfavorable ratio of breast size to tumor size and with certain tumor locations, before-mentioned problems can occur more frequently after breast-conserving therapy. Loss of volume in the affected breast, excess skin, resulting dislocation of the nipple–areola complex (NAC) and asymmetries to the opposite side can impair the aesthetic result. In particular, the dislocation of the NAC from the midline is perceived as disturbing. Especially after resection of breast tumors in the lateral breasts, the lateral “pillar” of the NAC can be weakened, and scar contracture of the skin or within the glandular tissue can dislocate the NAC from the midline (Figure 1).

Oncoplastic surgery can reduce these problems [5]. In addition to the classic oncoplastic reduction mastopexy, a local flap from the breast itself or the thoracic wall is a good alternative [5]. Classic flaps are the axial, thoraco-epigastric flap from below the infra-mammary fold (IMF) of the breast and its perforator-based version as AICAP (anterior intercostal artery perforator), and the laterally located perforator flaps such as the LICAP flaps (lateral intercostal artery perforator). Other perforator flaps described are based on branches of the lateral thoracic artery and the thoracodorsal artery [6,7,8,9]. These flaps can be used with skin, if required, or buried without skin. Large axially pedicled rotational flaps and bi-lobed flaps are described for larger defects [10].

A classic flap that has so far only been used in special case reports and very rarely in cancer surgery on the breast and axilla is the local, pedicled rhombus-shaped flap [11,12,13,14]. First described by Limberg in 1950 and later modified by Dufourmentel in 1962, it has a wide range of applications all over the body [15,16,17]. The flap was popularized in the 1970s for defects after skin cancer and in hand surgery. In the modified version of Dufourmentel, the flap itself is narrower and, thus, smaller, and the flap base is wider than in the Limberg flap. Currently, both flaps are used in defect closure after excisions in the axilla, pilonidal sinus or after skin cancer, to name just a few indications. One advantage of the rhombic flaps is that they have a safe 1:1 length-to-width ratio and axial, randomized blood supply [18]. This means that there is no need to search for a perforator that perfuses the flap, and flap harvest is simple. The donor area of the flap can be closed primarily. The only prerequisite is that the skin at the donor site shows some laxity, like is often the case at the breast and chest. The flap does leave a scar in the donor site, however, a smaller one compared to other flaps.

On the breast, the rhombus-shaped flap can be used as an all-layer flap to cover defects. The flap can be harvested inferiorly and laterally on the breast and can close all-layer defects on the breast. However, it can also be deepithelialized and buried for volume defects of the lateral breast to fill the parenchymal defect of the breast after tumor removal. This technique has not been described yet. In the axilla, it can be used to cover full-thickness defects after resection of recurrences when skin is involved.

We aim to further define the technique, describe the theory of the flap, and illustrate the application of the flap.

## 2. Surgical Technique

The defect should be planned in a rhombus shape. The sides of the rhombus are of equal length and ideally have an angle of 60° and 120°. A rhombus also has perpendicular diagonals. In the case of the Limberg flap, the flap that closes the defect is planned as an extension of the short diagonal (halves the two 120° angles). The distance corresponds to the length of the diagonal. The second incision of the flap is placed according to the defect angle of 60° so that it runs parallel to the defect at the same length. This creates a second rhombus, the fourth side of which is not incised, however, as it represents the flap pedicle (Figure 2). The flap is usually harvested as an all-layered, subcutaneous flap and is, thus, randomized via the pedicle and axially perfused. Length and width correspond to each other, so that the flap has a 1:1 ratio and can, therefore, be classified as being perfusion-safe. The flap is transposed into the defect, and the donor area is primarily closed in two layers.

If possible, the flap should be planned in such a way that (1) the tissue in the donor area is soft and elastic, (2) the scar in the donor site is either in the RSTL (relaxed skin tension lines) or can be concealed and (3) the blood supply to the flap is not restricted, e.g., by other close incisions for sentinel lymph node removal, which could disrupt blood flow to the flap pedicle.

The flap is designed with a length-to-width ratio of 1:1. Therefore, complications from impaired perfusion are unlikely if the described planning details are adhered to. Care must be taken not to plan for the flap pedicle to be too close to other incisions, as the flap is mostly axially perfused. Anatomically, in some regions of the breast and chest wall, the flap could also be harvested as a perforator flap. However, this counteracts the idea of a simple flap harvest. When the deepithelialized buried flap is sutured into the defect, the sutures should be applied without tension to avoid any fat necrosis. 

Theoretically, the flap can be harvested 90° around the defect from four different areas. For lateral breast defects, the flap can be pedicled from the upper or inferior side for inferior breast defects from the medial or lateral side. Theoretically, larger defects can be shaped as 2–3 rhombi and then closed with two or three Limberg flaps (Figure 3).

## 3. Results

We have been using the rhombic flap as a routine procedure mainly for lateral breast tumors and axilla since we introduced the flap into our technical repertoire more than 20 years ago. We have noted that this easy technique is not widely known and used despite its obvious advantages. Therefore, typical examples are used to illustrate the versatility of this flap.

### 3.1. Breast Cancer with Skin Resection and Local Replacement from the Chest

A 72-year-old patient with bilateral invasive breast cancer had a mastectomy and defect closure with a latissimus dorsi muscle flap on the right side. After breast-conserving therapy on the left side, she developed a recurrence in the upper lateral side of the left breast. Otherwise, she was stable regarding her disease. The tumor was close to the skin and measured 4 × 3 cm, and the resection of the skin was, therefore, indicated. Simple excision with primary closure at this lateral location would have carried the risk of NAC distortion; the oncoplastic volume and skin replacement from adjacent tissue were essential.

The resulting defect from tumor resection is planned as a rhombus (Figure 4). The flap is located within an area of excess skin, and the donor site can, therefore, easily be closed primarily. The resulting scars at the donor site will be less noticeable later, as they are located within the RSTL of the chest. Unlike in perforator flaps from the lateral intercostal artery system (LICAP), which represent a surgical alternative in this case, no perforators need to be identified in advance or dissected during surgery, and microsurgical expertise is not required. As a flap with a 1:1 length to width ratio, it is safely perfused from the dorsal side. As a result, the procedure is quick—in this case, the operating time was about 30 min—and can also be performed under local anesthesia, if necessary.

### 3.2. Axilla

A 62-year-old patient with a history of invasive breast cancer on the left side presented with an axillary recurrence 9 years after initial diagnosis. A core biopsy revealed the cancer diagnosis. The skin over the recurrence was retracted and involved (Figure 5). A PET/CT scan was negative for further metastasis. An all-layer rhombus-shaped resection, additional lymphatic dissection and defect coverage with a dorsally pedicled rhombus-shaped flap according to Limberg were performed. The donor site was primarily closed without tension using intracutaneous running closure with Monocryl 4-0 (Ethicon, Zug, Switzerland). Surgery was performed on an outpatient basis.

### 3.3. Breast Cancer Resection and Volume Displacement with a Buried Flap

In this 53-year-old patient, a mammogram and MRI showed a 3.5 cm × 4 cm tumor with little satellites in the right breast. A core biopsy revealed invasive breast cancer. As the breast was very firm and the relation of breast to tumor size was not ideal, a volume displacement was planned. In this case, the skin does not need to be resected, and the incision was placed right above the tumor (Figure 6). The removed tumor from the upper lateral breast weighs 85 g; in a moderately large breast, this correlates to about 1/5 to 1/4 of the total breast volume. To fill the dead space and relocate volume, a Limberg flap was planned. The flap is pedicled dorsally and cranially and incised in all layers. The flap is then temporarily fitted into the defect, and the patient is seated. This allows the assessment of volume and the NAC position. The part of the flap that will be buried is marked. The inferior, buried portion of the flap is then deepithelialized and sutured into the cavity that resulted from the tumor resection. No need for suction drains. Breast surgery was combined with a sentinel lymph node resection. Pathology revealed an invasive breast cancer and concomitant DCIS. Margins were clear, and the patient received subsequent hypofractionated radiotherapy of the breast with 40.05 Gy (15 × 2.67 Gy).

## 4. Discussion

In breast-conserving surgery, (1) the tumor should be safely removed oncologically without residual tumor, (2) the breast should heal primarily without pronounced seroma/fat necrosis so that adjuvant radiotherapy can take place without delay and with a low complication rate, (3) the breast should look aesthetically pleasing (no dislocation of the NAC, preserved volume in the quadrants) and (4) be symmetrical to the opposite side. At first, BCT is considered to be an oncological intervention, but afterwards, it is judged by its cosmetic result. 

Almost every breast-conserving operation that does not necessarily resect skin results in a disproportion of skin and volume. In small resection volumes and tight, elastic skin, this can be compensated for by mobilizing the skin and its laxity. In cases of larger resections and, in correlation, small breast volumes, ptotic breasts and unfavorable tumor localization, conventional breast-conserving therapy reaches its cosmetic limits. If only parenchymal tissue from the breast is used for reconstruction, this can lead to excess skin, loss of volume, dislocation of the NAC and loss of symmetry. Dead space and perfusion problems within the parenchymal flaps can cause seromas and fat necrosis, which can be exacerbated by radiotherapy. Aesthetic impairments and pain can result, further affecting patients in this already difficult phase of the disease [2,3,4].

Oncoplastic techniques and extreme oncoplasty can reduce these problems [19]. The domain of oncoplastic surgery is the volume displacement technique for larger and/or ptotic breasts. This involves using mastopexy and reduction techniques to close the tumor-related cavity through the available breast tissue, reshaping the breast volume and redraping the excess skin. As the NAC is transposed to a determined position, this normally prevents displacement of the NAC. Additionally, excess skin is controlled by resection in aesthetic sections [20]. However, it leads to more scars, namely circumareolar, vertical and usually in the infra-mammary fold. In patients with comorbidities such as hypertension, diabetes and other risk factors, the risk of major complications from surgery is increased [21]. In addition, the opposite breast must always be adjusted to maintain symmetry. 

In the volume replacement technique, the volume resected due to the tumor is replaced from outside the breast [10,19]. The most extensive operation is the replacement of the gland or large parts of it using a free flap [22].

For limited defects, it is also possible to replace the volume and fill the cavity created by the tumor resection using a local flap. Local flaps can be taken from the epigastrium, the lateral thoracic wall and the back. Standard flaps are the perforator-based flaps from the intercostal system [8,9,22,23]. The advantage of these flaps is that they are mobile and can be transposed around the perforator vessels [24]. The disadvantage, however, is that the perforators must first be identified using at least a Doppler or ultrasound examination, and flap dissection requires microsurgical expertise. Additionally, arterial and/or venous perfusion might not be sufficient to ensure flap viability in all flap regions, especially in the distant parts. The complication rate for these flaps is high, with an overall complication rate between 7% and 37.8% and a reoperation rate of 11% [6,9,25]. In case complications occur, adjuvant radiotherapy has to be postponed [26]. When these flaps are buried, fat necrosis that occurs due to reduced perfusion or venous congestion is not visible or initially recognized but can manifest itself after radiotherapy. Therefore, the complication rate might be underestimated.

The advantages of the rhombus-shaped flap are its safety and simplicity, thus reducing the complexity of oncoplastic surgery [27]. It can be used wherever tissue is elastic enough to primarily close the donor area. In the face, trunk and buttocks, it is described for defect coverage. Also, after defects in the axilla from resection of hidradenitis suppurative, the Limberg flap is widely used [15,16,17,18]. On the buttocks, together with its modification of Dufourmentel, it is a standard flap for defect closure after pilonidal sinus resection [18,28].

However, the Limberg flap is not very well known in breast surgery. The existing literature describes only case reports on the closure of all-layered mastectomy or lumpectomy defects after cancer and after infections of the breast [11,12,13,29,30]. In 2007, da Silva Neto et al. wrote a letter describing their experience with an all-layered Limberg-flap with skin for lumpectomies in 250 patients over a period of almost 20 years in Brazil [31]. Due to the large defects, however, contralateral symmetrization was necessary. Complications are not described. A retrospective study from India in 2022 described 17 patients with T1-T2 tumors in the lower breast in whom an all-layered Limberg flap was used [32]. No complications or deformities of the breast occurred in any case. The technique we describe here using the deepithelialized flap for volume replacement has not been reported yet for this flap.

As an axial, randomized flap with a 1:1 ratio of width to length, it is almost always safely perfused. No microsurgical skills are required for preparation. The flap can be trimmed to fit into the defect; if the defect is rather round, the flap can still be harvested as a rhombus in order to be able to close the donor site well primarily, but subsequently adapted to the defect [33]. The flap base can be planned in 90° steps around the defect, depending on where the donor area (resulting scar) can be better closed or the resulting scar can be covered. Additionally, the positioning of the rhombus, which represents the defect, can usually also be planned flexibly, allowing the flap to be in an opportune area. It is important to harvest the flap from an area that is mobile and elastic so that the donor site can be closed well and that the resulting scar of the primary closure is planned in a manner that it either lies in the RSTL or can be concealed by a bra or the ptotic breast. 

We have not seen any healing problems or flap necrosis in our patients during the post-operative follow-up. Both flap types, the all-layered flaps with skin and the buried, deepithelialized flaps, have healed primarily in all cases. This matches the results of Bhargavan et al. [32]. The buried flap appears firmer in the first weeks compared to a flap with skin, but after 5–6 weeks, all flaps are soft. Adjuvant cancer treatment was not postponed in our patients. It is important to point out to patients in advance that a longer scar will remain with all-layered defects than after a simple excision. In rare cases, dog ears requiring correction may remain at the ends of the scar. They can be corrected under local anesthesia, if desired.

When harvesting the flap from the chest wall as a volume replacement, the maximum size of the flap that can be harvested depends on the possible primary closure of the donor site. This can be tested by pinching the skin. If the flap is used as a volume displacement within the breast, the maximum flap size depends on the breast size. A larger breast allows for a larger flap. However, we have not yet harvested this flap within a breast larger than 7 × 7 cm. The largest tumor resection that was filled with a buried Limberg flap weighed 130 g.

The disadvantage compared to perforator-based flaps is that the rhombus-shaped flap has a broad base and, therefore, cannot be flexibly rotated around its own axis, making it less mobile. On the other hand, it can be planned at any point (see Figure 3) and is not dependent on sufficient perforator vessels. 

Like the perforator-based local flaps, the rhombus-shaped flap can replace resected skin on the breast. This may be necessary for tumors that involve the overlying skin or show close proximity to the skin. The flap can also be used for secondary necrosis or local recurrence, and it is useful in axillary procedures that involve removal of axillary skin to avoid contractures and movement restrictions. 

As in standard breast conservation therapy, skin excision is usually not indicated, the flap can simply be deepithelialized and sutured into the resulting defect, and the dead space can be filled out. The flap is sutured into the cavity in such a way that the cavity is filled. In our setting, drains were mostly unnecessary. We have not seen any necrosis or infections with the flaps so far.

## 5. Conclusions

The rhombic-shaped flap is an easy and safe technique to provide volume after a larger tumor resection, especially in smaller breasts in the lateral and lower parts. It avoids dead space and prevents distortion of the NAC. Further, it can be used to close defects if the skin has to be resected. Symmetrization surgery is rarely necessary. The scars correspond to those of other oncoplastic flaps but are less than in oncoplastic reduction/mastopexy techniques.

## Figures and Tables

**Figure 1 cancers-16-03101-f001:**
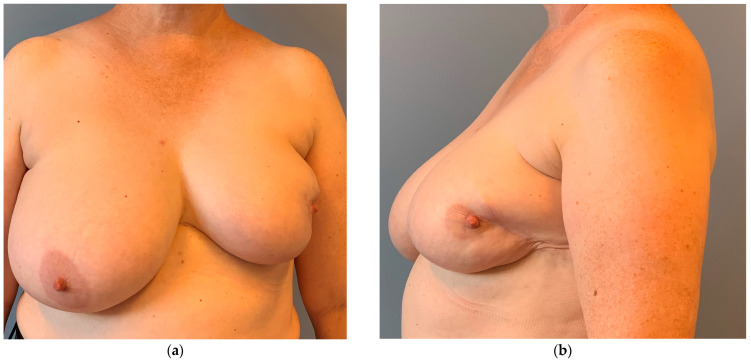
Unfavorable cosmetic result 2 years after externally conducted breast-conserving therapy on the left breast and adjuvant radiation therapy. (**a**) The nipple–areola complex (NAC) is dislocated from the midline. (**b**) Volume loss and scarring further impair the cosmetic result. A primary local flap for volume replacement or oncoplastic reduction mastopexy would have been a better choice.

**Figure 2 cancers-16-03101-f002:**
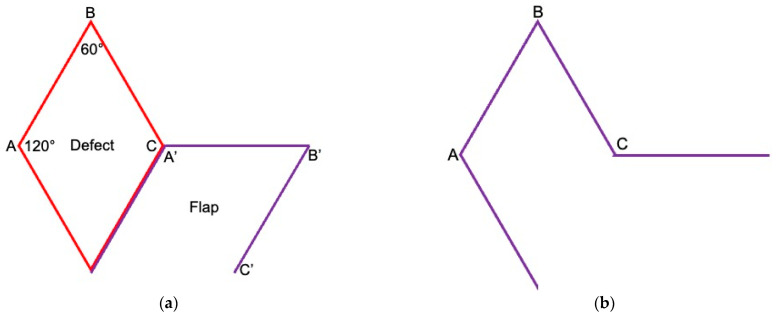
Principle of the rhombic flap according to Limberg. (**a**) The defect (red) is shaped like a rhombus. The sides of the rhombus are of equal length and have an ideal angle of 60° and 120°. The flap (purple) has the same measurements as the defect. The flap that closes the defect is planned as an extension of the short diagonal (halves the two 120° angles, here A to C = A′ to B′). The second incision of the flap, B′ to C′, runs parallel to the defect, again with the same length. The base of the flap is not incised. The flap is transposed into the defect so that A′ is sutured to A, B′ to B and C′ to C. (**b**) The shape of the final scar after the defect and the donor site of the flap are closed.

**Figure 3 cancers-16-03101-f003:**
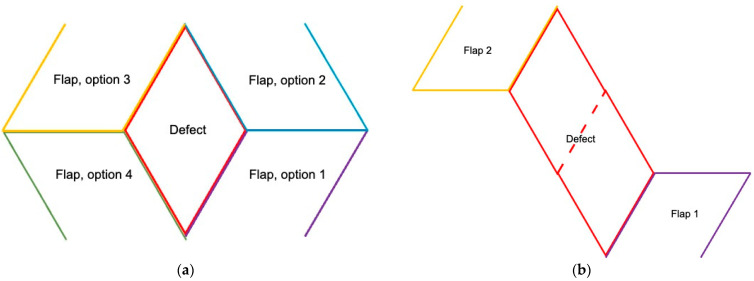
(**a**) The exact planning of the position (axis) of the rhombic defect usually has some play. Additionally, the definitive flap can be harvested 90° around the defect; thus, four options exist to close the defect, here named option 1 to 4. The best flap would be the one that adds the same amount of volume and whose donor site is best concealed, or whose donor site skin is most elastic to facilitate primary closure. (**b**) A larger defect (red) can be divided into two rhombi (dotted line). Each of the two smaller rhombi is closed by a separate Limberg flap (purple flap 1 and yellow flap 2). Even larger defects can be divided into 3 or 4 rhomboids. Caution: The primary closure of 4 Limberg flaps that are not buried results in a swastika as a scar at the end.

**Figure 4 cancers-16-03101-f004:**
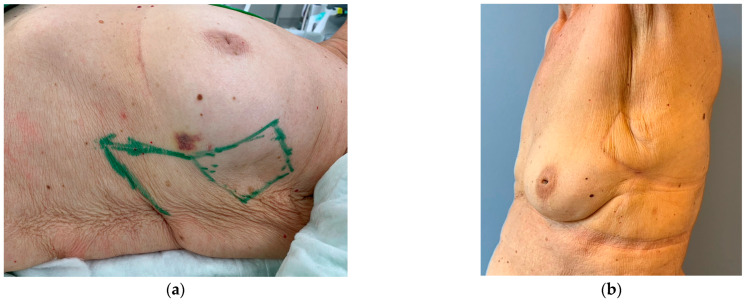
Patient with bilateral invasive breast cancer; recurrence in the upper lateral side of the left breast. (**a**) The all-layered tumor resection was planned in a rhombic shape. The flap was pedicled dorsally and inferiorly with the donor site in an area of excess skin. Alternative pedicles would have been placed dorsally and cranially, or cranially and anteriorly. However, by placing it dorsally and inferiorly, the area of skin excess was used best with direct closure of the donor site. (**b**) Result after 4 months. No dislocation of the NAC, no visible volume loss of the breast.

**Figure 5 cancers-16-03101-f005:**
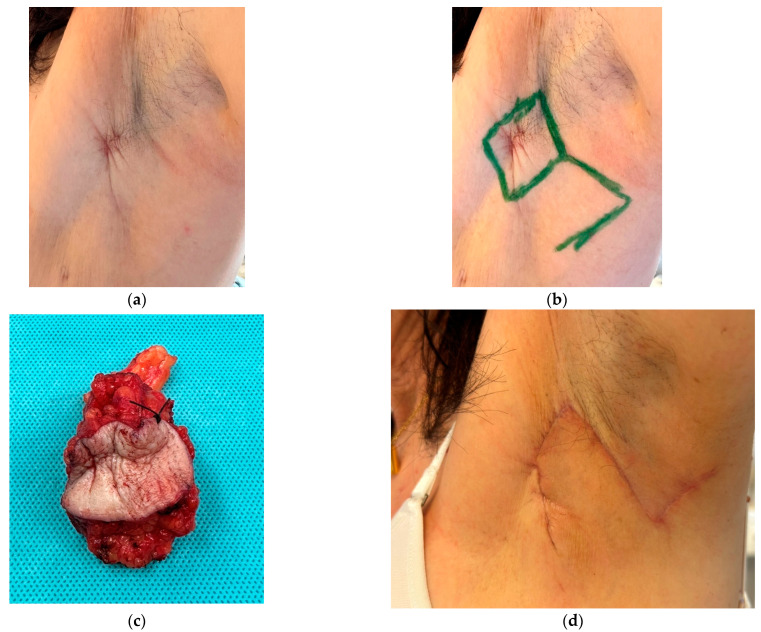
Patient with axillary recurrence on the left axilla 9 years after initial invasive breast cancer diagnosis: (**a**) Involvement of the skin is clearly visible. A resection with primary closure will result in a contracture and limitation of movement of the arm. (**b**) The resection was planned in a rhombus shape. The flap was pedicled inferiorly. There is no need for “complicated” flaps. Larger resections and defect coverage can also be performed with a rhombus-shaped flap. (**c**) Resected axillary recurrence with skin. (**d**) 3 months after surgery with free margins: primary healing of the flap. No contracture in the axilla, and no impaired range of movement. The patient then received radiotherapy and adjuvant immunotherapy.

**Figure 6 cancers-16-03101-f006:**
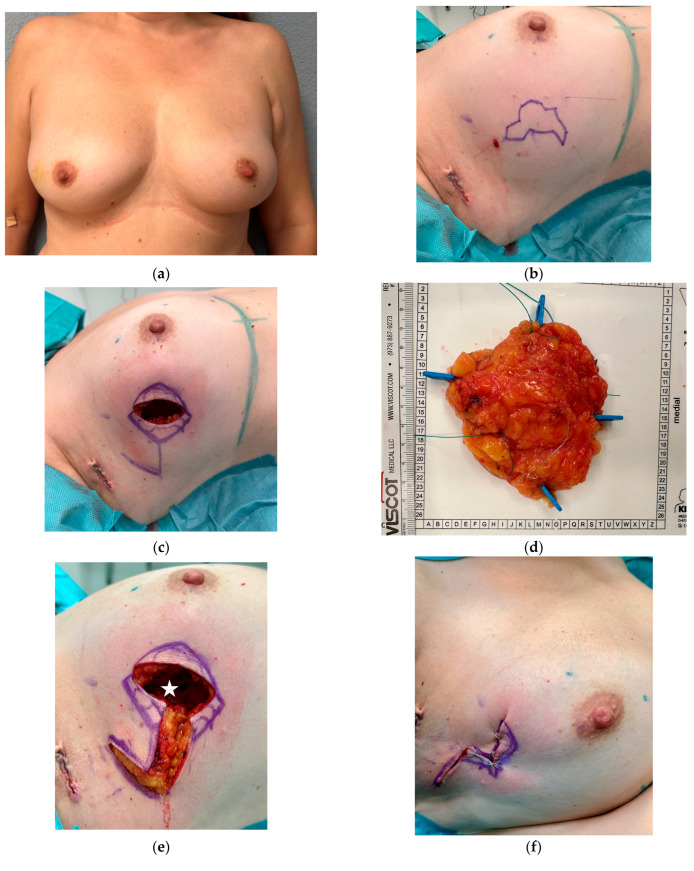
53-year-old patient with invasive breast cancer on the right breast. (**a**) Preoperative view with firm breast. (**b**) Wire marking and delineation of tumor borders. Tumor size: 3.5 × 4 cm. (**c**) The tumor is resected via a direct incision in this case. No skin resection is necessary. The flap is planned as a rhombus. The pedicle is placed cranially and dorsally. (**d**) Resected specimen; tumor weight is 85 g. (**e**) After tumor resection, the flap is incised in all layers. White star indicates the parenchymal defect after tumor resection. (**f**) Volume and NAC position are evaluated in the seating position with the flap temporarily fitted. Marking of the buried area of the flap. (**g**) Deepithelialization of the inferior, buried part of the flap. Fixation of the flap with Vicryl 3-0. (**h**) Immediately after wound closure, using Monocryl 4-0 intracutaneously. Drains are not necessary. (**i**,**j**) illustrate two weeks after surgery. Volume appears symmetrical, and NAC is in the right position, despite the large resection. (**k**,**l**) show results four months after surgery and eight weeks after radiation therapy. The ultrasound (**k**) shows no dead space/cavity or necrosis. The flap has healed well. The white arrows show the anterior deepithelialized buried part of the flap. The frontal view shows a symmetrical result and no NAC dislocation after radiation therapy.

## Data Availability

No new data were created.

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
