# Peer review of "The Local Rhombus-Shaped Flap—An Easy and Reliable Technique for Oncoplastic Breast Cancer Surgery and Defect Closure in Breast and Axilla"

_cancers, 2024, doi:10.3390/cancers16173101_

Round 1

Reviewer 1 Report

Comments and Suggestions for Authors

Dear Authors, Your article is very well written and in a very clear fashion, congratulations on that matter! However, it refers to  the "revival" of a surgical technique and, therefore, I would have enjoyed seeing some statistic data from you own collection of cases,  because I think that ,maybe some retrospective data would have offered a greater and more precise insight into the advantages and disadvantages of this technique, much more than a few examples of cases are. I will recommend the publication of this article in its current form, because I like it and I have enjoyed reading it, although it does not describe the precise characteristics of a batch of patients(operated with this technique), but mostly it depicts different traits from a few patients, and nothing more. Nevertheless and regardless of these previous sentences, in my opinion , articles such as this one should be published  more because they are discussing a technique with its pros and cos and all surgeons would potentially benefit from the information offered. All the best, respectfully yours, a Reviewer. 

Author Response

Reviewer 1

Comment 1: Dear Authors, Your article is very well written and in a very clear fashion, congratulations on that matter! However, it refers to  the "revival" of a surgical technique and, therefore, I would have enjoyed seeing some statistic data from you own collection of cases,  because I think that ,maybe some retrospective data would have offered a greater and more precise insight into the advantages and disadvantages of this technique, much more than a few examples of cases are.

Response: Thank your positive review. I understand the idea of a retrospective analysis. However, a retrospective evaluation does not offer new information. The flap is safe and we do not see any problems, if properly executed. We intended to describe the technique in an instructional way to allow other breast surgeons to use this flap. 

Comment 2: I will recommend the publication of this article in its current form, because I like it and I have enjoyed reading it, although it does not describe the precise characteristics of a batch of patients(operated with this technique), but mostly it depicts different traits from a few patients, and nothing more. Nevertheless and regardless of these previous sentences, in my opinion , articles such as this one should be published  more because they are discussing a technique with its pros and cos and all surgeons would potentially benefit from the information offered. All the best, respectfully yours, a Reviewer.

Response 2. Thank you for your assessment. We have added further details regarding the technique and possible complications.

Reviewer 2 Report

Comments and Suggestions for Authors

Dear Editor and Authors,

Thank you for the opportunity to review the manuscript entitled “The local rhombus-shaped flap – an easy and reliable technique for oncoplastic breast cancer surgery and defect closure in breast and axilla”. The authors described their technique of using Limberg flap in breast conserving surgeries due to cancer, presented three cases and discussed the utility of the technique. They concluded that the advantages of the rhombic flap are its safety and simplicity to add volume and close defects, thus reducing the complexity of oncoplastic surgery. I consider the paper appropriate for the Journal /especially for oncologic/breast surgeons/ and worth publishing in this Journal. 

However, I have some remarks:

-       I am not sure about the type of the article and if it fits the Journal requirements, this is not an original research manuscript not a review. Did the Authors intend to present it as Case report? /is such type accepted in the Journal?/ - the solution could be to rewrite the paper as original. You said :“We have been using the rhombic flap as a routine procedure mainly for lateral breast tumors and axilla since we have introduced the flap into our technical repertoire more than 20 years ago.” If so – you could do a retrospective analysis of your results /how many cases?/ – analyze sized of tumors, complications, aesthetic results, patients’ satisfaction, and rewrite the paper as “original research”. The presented cases can be kept and shown on photos as examples for different /but I guess representative/ uses of the flap.

-       The aim should be clearly stated 

-       Abstract is not informative in its current form – it should refer to the type of the paper

-       Add relevant references, e.g :

Bhargavan RV, Augustine P, Cherian K. Limberg Flap in Breast Oncoplasty for Carcinoma Breast Revisited-a Tertiary Cancer Centre Experience. Indian J Surg Oncol. 2022 Dec;13(4):876-879. doi: 10.1007/s13193-022-01589-5.

Tanaka, S., Nohara, T., Nakatani, S. et al. Esthetic result of rhomboid flap repair after breast-conserving surgery for lower quadrant breast cancer lesion with skin invasion: Report of two cases. Surg Today 41, 832–836 (2011). https://doi.org/10.1007/s00595-010-4355-4

Manabu Watanabe, Jun Morioka, Two Cases of Breast Reconstruction with a Limberg Flap, Nihon Gekakei Rengo Gakkaishi (Journal of Japanese College of Surgeons), 2016, Volume 41, Issue 4, Pages 567-571, Released on J-STAGE August 30, 2017, Online ISSN 1882-9112, Print ISSN 0385-7883, https://doi.org/10.4030/jjcs.41.567,

-       Conclusions in their current form are adequate for original research results 

Author Response

Reviewer 2

Comment 1: I am not sure about the type of the article and if it fits the Journal requirements, this is not an original research manuscript not a review. Did the Authors intend to present it as Case report? /is such type accepted in the Journal?/ - the solution could be to rewrite the paper as original. You said :“We have been using the rhombic flap as a routine procedure mainly for lateral breast tumors and axilla since we have introduced the flap into our technical repertoire more than 20 years ago.” If so – you could do a retrospective analysis of your results /how many cases?/ – analyze sized of tumors, complications, aesthetic results, patients’ satisfaction, and rewrite the paper as “original research”. The presented cases can be kept and shown on photos as examples for different /but I guess representative/ uses of the flap.

Reply:  Thank you for your accurate review. We intended to present the paper as a "surgical instruction". We aimed to describe and show a simple technique that is not widely used. The technique to bury the Limberg flap was not described elsewhere. Focus is on the technique to enable other surgeons to use the flap. Therefore, we have considered the paper as an "instructional article" or "innovations". However, the final decision should be made by the editors.

From our point a retrospective analysis does not add much new information, as the flap ist very safe and we do not see complications or cosmetic problems from the flap. 

Comment 2: Abstract is not informative in its current form – it should refer to the type of the paper

Reply: We rewrote the abstract to give more informative details on the technique. 

Comment 3: Add relevant references, e.g :.....

Reply: Thank you for the additional references. Apparently, these were missed in our pub med search. The papers also highlight that the Limberg-flap for breast surgery is a good additional tool. We analyzed the papers and added the references. Additionally, we discussed their findings in the discussion section. Thank you for this valuable support. 

Reviewer 3 Report

Comments and Suggestions for Authors

Thank you for submitting your manuscript, “The local rhombus-shaped flap – an easy and reliable technique for oncoplastic breast cancer surgery and defect closure in the breast and axilla,” to Cancers.

Your study focuses on a very interesting topic, which could represent a useful tool for treating certain patients affected by breast cancer who are not suitable for common pedicle flaps.

Nevertheless, I recommend a minor revision prior to publication:

  • Could the authors describe whether they noticed any complications during the follow-up of the patients?
  • Could the authors provide their opinion on the limitations of this technique in terms of the volume of excised parenchyma that can be replaced and the extent of the skin island that can be drawn?

Your study has the potential to make a significant contribution to the literature on optimizing breast cancer treatment. With the above revisions, I believe your manuscript will be strengthened.

Thank you again for the opportunity to review your work. I hope you find these comments constructive.

Sincerely

Comments on the Quality of English Language

There are minor grammar errors throughout the manuscript. I recommend having a native English speaker or professional editing service review your manuscript for grammar and syntax. This will help refine the overall writing

Author Response

Comment 1: Could the authors describe whether they noticed any complications during the follow-up of the patients?

Reply: Thank you for your careful review and kind assessment. The flap is very robust and safe. We do not see any complications from the flap when executed properly. We have added more information to the paper and further described the technique to avoid any pitfalls. 

Comment 2: Could the authors provide their opinion on the limitations of this technique in terms of the volume of excised parenchyma that can be replaced and the extent of the skin island that can be drawn. 

Reply: We added the limitations in terms of size of the flap to the text. The size when taken from the chest wall is limited by the donor site. We can pinch the skin to assess the amount/size of flap that allows a primary closure of the donor site. The size of the flap when taken from the breast itself depends on the breast size. The example in the paper was a tumor resection of 85g. We have used the flap in resection volumes of 130g. The largest flap was 7x7cm. 

Comment 3. Your study has the potential to make a significant contribution to the literature on optimizing breast cancer treatment. With the above revisions, I believe your manuscript will be strengthened.

Reply: Thank you again!

Reviewer 4 Report

Comments and Suggestions for Authors

Dear Editor and Authors,

Thank you for asking me to evaluate this manuscript titled “The local rhombus-shaped flap – an easy and reliable technique for oncoplastic breast cancer surgery and defect closure in breast and axilla” by Drs. Hisham Fansa and Sora Linder from Zürich, Switzerland.

This is an very interesting and well written overview of a seldomly used technique, the pedicled rhombus-shaped flap for oncoplastic breast cancer surgery, which can offer a good cosmetic effect in a variety of cases.

This is a well presented work, it is thorough and well written and also nicely illustrated. The authors are concise but also descriptive to the extend needed to give the reader a good grasp of the technique. The use of specific case examples enhanced the description and were very informative. The discussion is also well structured and appropriately gives an overview of the benefits and challenges of the technique.

In conclusion, I feel the manuscript is quite good and acceptable for publication.

Kind regards to all.

Comments on the Quality of English Language

Minor, if any editing is needed. Language is good.

Author Response

Reviewer 4

Comment 1: This is a well presented work, it is thorough and well written and also nicely illustrated. The authors are concise but also descriptive to the extend needed to give the reader a good grasp of the technique. The use of specific case examples enhanced the description and were very informative. The discussion is also well structured and appropriately gives an overview of the benefits and challenges of the technique.

Reply: Thank you for your thorough review und kind assessment! 

Reviewer 5 Report

Comments and Suggestions for Authors

The manuscript presented for review is an interesting report of using romboid flaps for closing defects resulting after breast cancer or axillary surgery. The technique is sound and can in fact be use in the proposed fashion with decent results. My opinion is however that most tumors situated in the external quadrants may be adequately resolved by an elliptical incision from NAC to external margin of breast which allows for safe removal of the tumor and ample access for Lymphadenectomy/sentinel lymph node. The results are estetic and do not interfere with ulterior treatment. Yet for selected cases romboid flaps may be an option worth exploring.

My suggestion for improvement of the manuscript are :

1. Please provide additional information on history and current usage of romboid flaps. 

2.Discuss technic pitfalls which may hinder a romboid flap.

2.Discuss possible complication after such flaps and how they may interfere with Oncologic treatment. 

3. Discuss reports of such flaps in beast cancer available in literature and corresponding Oncologic results. 

4. Minor English editing needed

5. Since this manuscript does not present any original research but rather a collection of clinical cases the manuscript should not be labeled as article but as case report. 

Comments on the Quality of English Language

Minor editing needed. 

Author Response

Reviewer 5

Comment 1:  The manuscript presented for review is an interesting report of using romboid flaps for closing defects resulting after breast cancer or axillary surgery. The technique is sound and can in fact be use in the proposed fashion with decent results. My opinion is however that most tumors situated in the external quadrants may be adequately resolved by an elliptical incision from NAC to external margin of breast which allows for safe removal of the tumor and ample access for Lymphadenectomy/sentinel lymph node. The results are estetic and do not interfere with ulterior treatment. Yet for selected cases romboid flaps may be an option worth exploring.

Reply: Thank you for your thorough review and beneficial comments. We are with you on that topic. The rhombic flap should be reserved for larger resections/resections with skin involvement which would impair the aesthetic result without an oncoplastic approach. 

Comment 2: Please provide additional information on history and current usage of romboid flaps. 

Reply: We have added information on the history and expanded information on the current use of rhombic flaps on the body and breast. 

Comment 3: Discuss technic pitfalls which may hinder a romboid flap. Discuss possible complication after such flaps and how they may interfere with Oncologic treatment.

Reply: We have added more technical details on the flap harvest and insertion which are the key points for a possible complication. Additionally, we have discussed complications and their interference with adjuvant treatment. However, the flap is very safe and if properly executed complications tend to be rare. 

Comment 4:  Since this manuscript does not present any original research but rather a collection of clinical cases the manuscript should not be labeled as article but as case report. 

Reply: The idea behind this paper was to demonstrate the technique behind this useful, rarely used flap. We aimed to offer a "surgical instruction article" to broaden the application of this technique. The technique of the buried Limberg-flap on the breast is new and not published yet. Therefore, we consider the paper as an article on a surgical technique. However, we hare happy to leave the decision to the editors.

Round 2

Reviewer 2 Report

Comments and Suggestions for Authors

Dear  Authors,

Thank you for revising your paper according to “some” of my comments. Although I think the technique you presented is interesting and worth sharing among Journal readers, I do not agree with your reply:

 “From our point a retrospective analysis does not add much new information, as the flap ist very safe and we do not see complications or cosmetic problems from the flap.”

There is a fundamental difference for a reader who would like to apply new technique if the authors have just “invented” something and used it a few times or if they are using the method for 20 years, performed e.g. 200 cases with a low rate of complications and good results in e.g. 90%... The fact that you ‘do not see complications” is not reliable until you present the true data. But, of course, as you mentioned “the final decision should be made by the editors.”

Reviewer 5 Report

Comments and Suggestions for Authors

The authors have adressed all my concerns adequately and the manuscript can be published in the current form. I congratulate the authors for their work in trying to improve surgical techniques.